# Decadal and long-term boreal soil carbon and nitrogen sequestration rates across a variety of ecosystems

K.L. Manies[1], J.W. Harden[1], C.C. Fuller[1], and M.R. Turetsky[2]

[1]US Geological Survey, Menlo Park, CA, USA
[2]University of Guelph, Guelph, Ontario, Canada

*Correspondence to*: K.L. Manies (kmanies@usgs.gov)

**Abstract**

Boreal soils play a critical role in the global carbon (C) cycle; therefore, it is important to understand the mechanisms that control soil C accumulation and loss for this region. Examining C & nitrogen (N) accumulation rates over decades to centuries may provide additional understanding of the dominant mechanisms for their storage, which can be masked by seasonal and interannual variability when investigated over the short-term. We examined longer-term accumulation rates, using $^{210}$Pb and $^{14}$C to date soil layers, for a wide variety of boreal ecosystems: a black spruce forest, a shrub ecosystem, a tussock grass ecosystem, a sedge dominated ecosystem, and a rich fen. All ecosystems had similar decadal C accumulation rates, averaging $84 \pm 42$ gC m$^{-2}$ yr$^{-1}$. Long-term (century) C accumulation rates were slower than decadal rates, averaging $14 \pm 5$ gC m$^{-2}$ yr$^{-1}$ for all ecosystems except the rich fen, for which the long-term C accumulation rates was more similar to decadal rates ($44 \pm 5$ gC m$^{-2}$ yr$^{-1}$ and $76 \pm 9$ gC m$^{-2}$ yr$^{-1}$, respectively). The rich fen also had the highest long-term N accumulation rates ($2.7$ gN m$^{-2}$ yr$^{-1}$). The lowest N accumulation rate, on both a decadal and long-term basis, was found in the black spruce forest ($0.2$ and $1.4$ gN m$^{-2}$ yr$^{-1}$, respectively). Our results suggest that the controls on long-term C and N cycling at the rich fen is fundamentally different from the other ecosystems, likely due to differences in the predominant drivers of nutrient cycling (oxygen availability, for C) and reduced amounts of disturbance by fire (for C and N). This result implies that most shifts in ecosystem vegetation across the boreal region, driven by either climate or succession, will not significantly impact regional C or N dynamics over years to decades. However, ecosystem transitions to or from a rich fen will promote significant shifts in soil C and N storage.

**1 Introduction**

High latitudes soils store 50 % of the global soil carbon (C) pool (Tarnocai et al., 2009; Davidson and Janssens, 2006) due largely to physical factors such as low soil temperatures and wet soil conditions. As a result, C losses are 30 generally smaller than C inputs, even over long timescales (Ovenden, 1990) when disturbances such as insects and fire are included. The majority of net C storage is in the form of thick (many cm to several meters deep) organic soils overlying the mineral soil component. Climate change is expected to impact the boreal region in many ways, including thawing of permafrost and reduced precipitation (Hinzman et al., 2005). These and other changes can alter the dominant vegetation types. If the factors that moderate C storage shift, it is likely that the balance of C 35 inputs and losses will also change, impacting the net C balance. Because of the high amount of C stored in boreal soils, changes in these C stocks can substantially affect the global C budget (Chapin et al., 2000).

Many studies have examined boreal N availability, mineralization rates, and their influence on C storage (for example, Keller et al., 2006; Bonan and Van Cleve, 1991; Gundale et al., 2014; Allison et al., 2010), yet boreal nitrogen (N) stocks are less well studied. It is known that boreal forests have large stocks of soil organic N 40 (Valentine, 2006), with peatland stocks comprising approximately 10-15 % of the global N pool (Loisel et al., 2014). The majority of N within boreal ecosystems resides within the organic and mineral soil (Merila et al., 2014). The size of these soil stocks changes with soil drainage and dominant vegetation (Van Cleve et al., 1983), in part as a result of N loss from fire (Harden et al., 2002). Understanding N stocks and availability is important because N controls many aspects of plant productivity and, therefore, cycling of C and N are closely linked (Vile et al., 2014).

Accumulation rates of C and N in soils vary according to the time scale, ecosystem type, and region studied. Short-term accumulation rates are higher than long-term rates because there is little influence of disturbance (Turunen et al., 2004). Accumulation rates also vary by ecosystem. Peatlands accumulate about 20-30 gC m$^{-2}$ yr$^{-1}$ over the long-term (see, for example, Yu et al., 2013; Jones and Yu, 2010; Turunen et al., 2002; Roulet, 2000), with bogs typically having higher rates of C accumulation than fens (Tolonen and Turunen, 1996). Long-term C 50 accumulation rates in peatlands are driven by growing season length and photosynthetically active radiation (PAR: Charman et al., 2013). Black spruce (*Picea mariana*) forests have C accumulation rates ranging between 10 – 40 gC m$^{-2}$ yr$^{-1}$ ( Harden et al., 2012; Trumbore and Harden, 1997; Goulden et al., 2011; Rapalee et al., 1998; Harden et al., 2000), depending on soil drainage class and timescale studied. C accumulation rates of these ecosystems are

related to fire and burial of C into deeper soil layers (O'Donnell et al., 2011; Harden et al., 2012). Accumulation rates of N in northern peatlands have been found to average from 0.5 – 0.9 g N m$^{-2}$ yr$^{-1}$ (Loisel et al., 2014; Wang et al., 2014), although this rate has changed over time. Fens have higher rates of N accumulation than bogs (Meng et al., 2014; Trumbore et al., 1999), reflecting the importance of plant and groundwater inputs to fen ecosystems. Little is known about C or N accumulation rates for the ecosystems other than peatlands or forests (i.e., shrubs, grass tussock, etc.) that characterize boreal landscapes. Although these other ecosystem types cover less area than black spruce forests and peatlands (DeWilde and Chapin, 2006), they still comprise an important part of the Interior Alaskan landscape.

Differences in the vegetation and environmental conditions among the varied ecosystems of Alaska influence their C & N accumulation rates. Litter production varies among vegetation (Camill et al., 2001), thereby impacting rates of C and N input to the soil. The chemical content and concentration of litter also varies among vegetation types (Hobbie, 1996). Litter composed of more complex C compounds and/or lower lignin:N ratios can have lower decomposition rates and, therefore, lower rates of C loss and relative N retention. Vegetation is also correlated with soil drainage (e.g., soil moisture; Camill, 1999), the presence of permafrost, and thus the thickness of insulating organic soil layers (Lawrence and Slater, 2008; Harden et al., 2000). All of these factors affect rates of decomposition (Rapalee et al., 1998; Wickland et al., 2010; Dioumaeva et al., 2002; Wickland and Neff, 2008; Treat et al., 2014), losses due to combustion (Harden et al., 2002), and rates of mineralization (Bonan and Van Cleve, 1991; Valentine, 2006), with wetter sites having lower rates of C and N loss. Because litter inputs, litter quality, the presence of permafrost, and soil moisture and temperature all affect rates of C and N accumulation and vary among ecosystem types, it follows that rates of C and N accumulation also vary according to ecosystem type, with ecosystems with more labile litter and/or with warmer soil temperatures storing less C and N over the long-term.

Many accumulation studies focus on daily, seasonal, or annual timescales. These studies use either chamber or eddy covariance techniques to measure net ecosystem exchange (NEE). These short-term investigations have led to insights regarding the importance of water table to the net C and N budget (Chivers et al., 2009; Ise et al., 2008), the role of shallow soil layers in trace gas emissions (Wickland et al., 2010), and the importance of seasonal variations to the annual net C balance for various boreal ecosystems (Euskirchen et al., 2014). Additional insights into the drivers of C and N storage can be obtained by examining accumulation rates over longer time frames, such

as decades or centuries. Through such investigations we have learned how C accumulation rates increase as soil moisture increases (Rapalee et al., 1998), how N deposition increases C accumulation rates (Turunen et al., 2004), and how disturbances, such as fire (Pitkanen et al., 1999), reduce C and N accumulation rates.

To help our understanding of longer term C and N accumulation rates in a variety of boreal ecosystems, we
compared soil-based C and N accumulation rates in five different ecosystems within Interior Alaska, each varying in soil moisture and dominant vegetation (black spruce, shrubs, tussock grass, sedge, or moss). These ecosystems were located along a moisture gradient, thereby controlling for factors such as parent material, climate, and topography, which influence soil formation (Jenny, 1941). We examined C and N accumulation rates on both decadal and century timescales to determine how the interaction of soil and vegetation influences these rates, and
thus, C and N storage over time. Based on differences in soil temperature, soil moisture, and litter quality, we predicted that the black spruce ecosystem would have the lowest rate of C and N accumulation while the rich fen would have the highest rate of C and N accumulation, with the values of the other ecosystem's accumulation rates residing somewhere in between.

**2 Methods**

Study sites were located within the Bonanza Creek Long-Term Ecological Research (LTER) site (64.70°N, 148.31°W), approximately 30 km south-west of Fairbanks, Alaska, within the floodplain of the Tanana River. This region of Interior Alaska is characterized by a mean annual temperature of -7 °C and mean annual precipitation of 300 mm (Hinzman et al., 2006). We studied soils in five ecosystems located along a ~300-m transect, each of which were dominated by a different type of vegetation. The ecosystems, presented in order as they appear on the
landscape, are: 1) a closed-canopy black spruce forest with a feathermoss and *Ericaceous* shrub understory (hereafter "black spruce"), 2) a shrub system comprised of willow (*Salix* sp.) and birch (*Betula* sp.) with an understory dominated by *Chamaedaphne calyculata* and sparse moss cover ("shrub"), 3) a tussock grass system dominated by *Calamagrostis canadensis* with some brown mosses present ("tussock grass"), 4) a peatland dominated by emergent vegetation such as *Equisetum fluviatile* ("sedge"), and 5) a moss dominated rich fen,
comprised of both brown mosses and *Sphagnum* sp. ("rich fen"). These ecosystems varied in moisture status related to water table and presence of permafrost (Table 1; Waldrop et al., 2012). This transect extends from the

toe slope of an adjacent upland forest into a ~1.8 km$^2$ fen complex. Although in the Tanana floodplain, the sites are ~1.5 km from the current location of the river and appear to be relatively stable since site initiation in 2005. These sites have also been a part of other studies, including examining controls on ecosystem respiration (McConnell et al., 2013), examining differences in the soil biotic community and their impact on soil C turnover (Waldrop et al., 2012), understanding how changing water table level impacts C cycling within the fen (Kane et al., 2013; Chivers et al., 2009), and using eddy covariance methods to calculate net ecosystem productivity (Euskirchen et al., 2014).

Three soil cores, encompassing all of the organic soil and extending into the mineral soil below, were collected at each site at randomly selected locations within an area less than ~10 m$^2$. Sampling for the black spruce and low shrub site occurred during the summer and samples were obtained using a combination of soil blocks cut to a known volume and using a 'Makita' coring device (4.8 cm diameter; Nalder and Wein, 1998). Soil cores from the other three sites were obtained in the spring, when the ground was frozen, using a SIPRE corer (7.6 cm diameter; Rand and Mellor, 1985). Each soil profile was then divided into subsamples representing soil horizons. Soil horizon thicknesses ranged between 2-14 cm, with 85% of samples having a thickness ≤5 cm. This separation occurred either in the field or, if frozen, in the lab, based on visual factors such as level of decomposition and root abundance. Each horizon sample was described using modified soil survey techniques (Manies et al., 2016).

Soils horizon samples were processed in several steps: first they were air dried (20-25 °C) and then homogenized. The samples were then split into two parts: an archive split and an analytical split. The analytical split was oven dried and then ground. Soils classified as organics were oven dried at 65 °C and ground to <0.25 mm using a Cyclone mill (Udy Corporation., Ft. Collins, Colorado). Mineral soils were oven dried at 105 °C and ground using a mortar and pestle until the soil passed through a 60 mesh (0.25 mm) screen. Total C and N content was analyzed using a Carlo Erba 1500 Series 2 elemental analyzer (Fisons Instruments; Manies et al., 2016). C and N stock inventories were calculated as the total amount of C or N within the profile to the organic/mineral soil boundary. Recent ages were determined by measuring $^{210}$Pb and $^{226}$Ra activities using gamma spectrometry by means of a Princeton Gamma HPGe germanium well detector using previously described methods (Van Metre and Fuller, 2009; Fuller et al., 1999). Total $^{210}$Pb activity was measured and is the combination of supported $^{210}$Pb (produced in situ through the decay of $^{226}$Ra in the soil) and unsupported $^{210}$Pb (produced in the atmosphere and added to the ecosystem through atmospheric deposition). Unsupported $^{210}$Pb was defined as the difference

between measured total $^{210}$Pb and $^{226}$Ra. Subsamples from each soil horizon within the profile, starting at the surface, were measured until unsupported $^{210}$Pb was not detected.

Unsupported $^{210}$Pb values were used to calculate dry mass accumulation rates (MAR, g cm$^{-2}$ yr$^{-1}$) for each soil horizon, from which dates of formation were calculated using both the Constant Flux: Constant Sedimentation method (CF:CS; Robbins, 1978) and Constant Rate of Supply method (CRS; Appleby and Oldfield, 1978). To account for compaction and loss of mass due to organic matter decomposition, both methods modelled unsupported $^{210}$Pb as a function of cumulative dry mass (g/cm$^2$), not depth (Appleby and Oldfield, 1992). Cumulative dry mass is the product of bulk density of the horizon (g/cm$^3$) and the horizon thickness (cm). The CF:CS method is based on fitting the decrease in unsupported $^{210}$Pb versus cumulative dry mass to a single exponential function based on decay, and thus, estimating an overall MAR by assuming a constant MAR through time. The CRS method assumes a constant rate of input of unsupported $^{210}$Pb activity per unit area and determines a mass accumulation rate for each soil horizon sampled by mass balance using the integrated unsupported activity of the whole profile and, thus, accounts for changes in MAR over time. The age of each sample interval is calculated from the resulting MAR from the surface downward. Uncertainty of the CRS MAR and resulting ages are derived from counting error, propagated from the top of the core downward (Binford, 1990; Van Metre and Fuller, 2009). As the soil profiles become deeper, and thus older, the total $^{210}$Pb activity approaches the supported activity, with the difference (unsupported activity) becoming similar to or less than the uncertainty in the measurement (which is propagated from the top of the core downward) (Binford, 1990; Van Metre and Fuller, 2009). At some point the magnitude of these errors become larger than the age estimated for that horizon (for example, the estimated age of the 19-22 cm horizon of BZBS 4 was 143 yrs old with an estimated error of 144 yrs; Table S1). This tends to occur for horizons dated older than 1920. To minimize these errors we constrained our decadal C accumulation rates to only include organic soil that had formed within the six decades previous to our sampling. Decadal C accumulation rates were calculated as the cumulative mass of C from the moss surface for the base of the that soil horizon, divided by the age of this soil horizon using the CRS age, which is more appropriate for ecosystems with variable rates of accumulation (Appleby and Oldfield, 1978; MacKenzie et al., 2011).

We also dated macrofossils, obtained from several processed, and therefore homogenized, soil horizons, using AMS radiocarbon measurements for comparison to $^{210}$Pb ages. (Suppl. Material S2). Additionally, bulk soil

samples, with roots removed, were submitted from the basal organic soil horizon to determine the timing of basal organic soil horizon formation. These samples were submitted to the USGS extraction laboratory (Reston, VA) for complete combustion and trapping of $CO_2$. Targets were prepared and submitted for accelerator mass spectrometry at Lawrence Livermore National Laboratory. Resulting $^{14}C$ data were corrected for $^{13}C$ and then

calibrated using CALIB v 7.0 (intercal13; Reimer et al., 2013), or, if they dated post-1950, CALIBomb (intercal13, NHZ1 curve extension). Long-term C accumulation rates were calculated as the amount of C within the organic soil profile divided by the $^{14}C$ age of that ecosystem. Ecosystem age was calculated as the average of the minimum and maximum $^{14}C$ calibrated ages (Suppl. Table S2).

**3 Results**

**3.1 Carbon, Nitrogen, and $^{210}Pb$ Inventories**

The rich fen site has significantly deeper organic soils than the other four sites (p<0.001), resulting in four or more times the amount of C and N than the other ecosystems (Table 2). Average unsupported $^{210}Pb$ inventories (dpm/cm$^2$) for each of the five ecosystem types were statistically similar (p=0.62, Table 2), which indicates that atmospheric input is the same for all ecosystem types and there are no apparent losses or transport of $^{210}Pb$

among sites. Whereas all the unsupported $^{210}Pb$ was found in organic soil in most systems, between 10-15% of the unsupported $^{210}Pb$ activity was found in the mineral soil horizons (2-4 cm thick horizons) for the tussock grass site. Because unsupported $^{210}Pb$ is deposited on the organic soil surface while bound to atmospheric aerosols and dust particles (Shotyk et al., 2015), we did not expect to find it in mineral soil layers. Its presence in mineral soil suggests that some of $^{210}Pb$ bearing particles may be transported downward in the grass ecosystem.  The potential

downward movement of unsupported $^{210}Pb$ would result in higher apparent CRS MAR and thus younger ages. Therefore, the tussock grass site was not included in the comparison of decadal accumulation rates.

**3.2 $^{14}C$ dates and dating methodology comparison**

$^{14}C$ dating of the basal organic soil layers provided information regarding the initiation of soil development. This approach shows that the rich fen is the oldest ecosystem, at approximately 1390 years old (Table S2). Age

estimates for the shrub and sedge ecosystems ranged between 700 and 856 yrs cal BP. Unfortunately, we did not

get ages for the black spruce or tussock grass ecosystem (due to sample size limitations). Therefore, for all ecosystems except the rich fen we used an initiation age of 780 yrs (the median of the two ages listed above). We justify this approach using the following logic. First, all of the ecosystems appear to be relatively stable and lay within ~300 m of each other, along an emergent landform that grades from the rich fen up to the black spruce

forest. Therefore, all ecosystems along this gradient likely formed within several hundred years of each other. This assumption is supported by the fact that the sedge ecosystem is only ~100 years older than the shrub ecosystem. The grass ecosystem also lies between the shrub and sedge ecosystems along the gradient; therefore, its age of formation is likely similar to the values measured for these two ecosystems. Although the black spruce ecosystem lies at the end of the gradient, a sensitivity analysis demonstrates that a dramatically different initiation age would

be needed to impact our results (Table S3). Therefore, even if 780 yrs is not accurate for the black spruce ecosystem, realistic variations in this value (+/- 400 years) would not change the outcome of our analyses.

For samples with both $^{14}$C and $^{210}$Pb, the ages defined by each technique were in general agreement (Fig. 1). We expected the $^{14}$C dates to lie somewhere within the $^{210}$Pb estimates due to the fact that the macrofossils were obtained from a homogenized sample comprised of the material from an entire soil horizon and so could have

formed at any time between when that soil horizon formed (the base) and the top of that horizon. In two instances the range of dates predicted using $^{14}$C was older than the $^{210}$Pb based age estimates (Fig. 1: Shrub, 8.5-12.5 cm; Rich fen, 5-10 cm). Because the 5-10 cm rich fen $^{14}$C date is also older than the two samples below it (10-15 and 15-20 cm), this $^{14}$C date is likely not accurate. The younger $^{210}$Pb date for the 8.5 – 12.5 cm Shrub-1 horizon could indicate that there has been some movement of $^{210}$Pb within the soil profile, which has been known to occur with

this dating technique (Turetsky et al., 2004). However, the $^{14}$C and $^{210}$Pb ages for the 4.5 – 8.5 cm horizon match well, which we would not expect if downward transport was a significant issue. In addition, adjusting our analyses to the $^{14}$C dates does not change our results. Therefore, we feel comfortable moving forward using the $^{210}$Pb age values.

**3.3 Decadal accumulation rates**

Decadal C accumulation rates (< 60 yrs) calculated from $^{210}$Pb CRS MAR were not statistically different among sites (Table 3; p-value=0.21), although the shrub ecosystem had the highest rate and the black spruce had the

lowest rate. Decadal rates ranged between 50 and 125 gC m$^{-2}$ yr$^{-1}$. Variability within each ecosystem type was high

(coefficient of variability: 12-60%). This variability is likely due to within-site heterogeneity, such as

microtopography, changes in vegetation, and differences in belowground biomass. Decadal accumulation rates of

the black spruce and rich fen ecosystems were similar to other literature values (Figure 2). N decadal accumulation

rates ranged from 1.4 to 5.6 gN m$^{-2}$ yr$^{-1}$ (Table 3). The black spruce ecosystem had significantly lower rates of N

accumulation than the sedge and the rich fen ecosystems (p=0.004). The rich fen rate had higher decadal N

accumulation rates (4.6 g m$^{-2}$ yr$^{-1}$) than values found for a Norwegian bog (0.6 - 2.1 g m$^{-2}$ yr$^{-1}$; Ohlson and Okland,

1998), but similar to rates found for a variety of fens (3.7 – 7.1 gN m$^{-2}$ yr$^{-1}$; Trumbore et al., 1999).

**3.4 Long-term accumulation rates**

Long-term rates of C accumulation ranged from 8 to 44 g C m$^{-2}$ yr$^{-1}$ across sites (Table 3). Variability was

highest in the grass tussock sites, which had a coefficient of variability of 65%, versus 12-34% for the other

ecosystems. Long-term rates of N accumulation ranged from 0.22 to 2.66 gN m$^{-2}$ yr$^{-1}$ (Table 3) with the black

spruce ecosystem having the lowest rate of long-term N accumulation. The shrub, tussock grass, and sedge

ecosystems had similar rates of long-term N accumulation. The rich fen had significantly higher rates of N

accumulation than the other ecosystems. The long-term N accumulation rate for the rich fen (2.66 gN m$^{-2}$ yr$^{-1}$) is

much higher than rates previously found for general peatlands (~0.5 gN m$^{-2}$ yr$^{-1}$; Loisel et al., 2014; Limpens et al.,

2006) and bogs (0.87 gN m$^{-2}$ yr$^{-1}$; Wang et al., 2014).

As expected, long-term C accumulation rates were lower than decadal rates for all ecosystems (Table 3; Fig. 2).

This decline in C accumulation rates is consistent with trends found in chronosequence studies using gas flux

(Baldocchi, 2008) and C stocks (Harden et al., 2012).  However, the difference between long- and decadal rates in

the rich fen was much smaller, indicating consistently high rates of C accumulation in this ecosystem (Table 3) and

suggesting some mechanism exists for preserving this C over longer time scales. Long-term C accumulation rates

for the rich fen are especially high compared to the other ecosystems (p<0.001), which were statistically similar

(Table 3). Our long-term C accumulation rates for the rich fen are similar to other rates based on changes in C stock

(Figure 2; Camill et al., 2009; Trumbore and Harden, 1997; Turunen et al., 2002).

**4 Discussion**

The ecosystems studied here have varied historically in their dominant vegetation, the presence or absence of permafrost, and hydrology. Despite these differences in ecosystem structure we found no significant differences in decadal rates of soil C accumulation (Table 3). Therefore, while inputs and losses of C into and from the soil system may vary across these ecosystems, the balance between inputs and losses for surface soil layers has been relatively similar across the past 60 years. McConnell et al. (2013) measured ecosystem respiration (ER) at the same five ecosystems and found higher ER in the grass and sedge ecosystems (see also Waldrop et al., 2012), with the other three ecosystems having similar, lower ER; thus the grass and sedge also have higher rates of net primary production (NPP) and generally cycle C more rapidly than the other systems. Across all ecosystem types, the shallow organic soil layers, which have been created in the past six decades, sequestered an average of 84 ± 42 gC $m^{-2}$ $yr^{-1}$.

Carbon inputs and losses also balance out similarly over the long-term (~1000 yrs) for all of the ecosystems we studied except the rich fen, which had greater long-term C accumulation rates than the other ecosystems (44 ± 5 gC $m^{-2}$ $yr^{-1}$; Table 3). The similarity in long-term C accumulation rates of the black spruce, shrub, grass, and sedge ecosystems (14 ± 5 gC $m^{-2}$ $yr^{-1}$) was initially surprising, as we expected the small, although not statistically significant, differences in the decadal C accumulation rates to add up over time, resulting in some significant differences in long-term accumulation. In hindsight, however, this result makes sense, as the total C stored in the organic soils of these four ecosystems are similar (Table 2). These results again demonstrate that even if the magnitude of C fluxes into or from the soils systems vary across these four sites, the overall balance between C inputs and losses are similar. We note that these four ecosystems fall along the same ER – soil temperature relationship (McConnell et al., 2013), suggesting that soil temperature may be one of the main drivers of C cycling for these sites.

Nitrogen accumulation rates have been studied much less frequently than rates of C accumulation. The long-term N accumulation rate for the rich fen in this study (2.66 g N $m^{-2}$ $yr^{-1}$) is five times higher than the 0.5 g N $m^{-2}$ $yr^{-}$ estimated by Loisel et al. (2014). There are several potential reasons for this discrepancy. First, Loisel et al. (2014) synthesized data from a wide range of peatland sites, including bogs, fens, and permafrost peatlands and thus included ecosystems with a broad spectrum of peat properties. In addition, Loisel et al. (2014) used time-

dependent C:N ratios of 65 and 40 to assign % N values for their soil horizons, resulting in average % N values that

never exceed 1.7 %. In contrast, the average % N value for our rich fen organic soil horizons was 2.4 %, resulting in

an average C:N ratio of 17 (Fig S1). In general, our results support Treat et al. (2015), who showed that fen C:N

ratios can be much lower than estimates used by Loisel et al. (2014), despite high variability (fen C:N averaging 29

+/- 15). Regardless, the amount of N within the rich fen ecosystem is relatively high.  Reasons for this high N

storage could include high rates of N inputs, either through high rates of biological $N_2$ fixation or through high N

concentrations in source water. The majority of studies on N fixation in peatlands have focused on *Sphagnum*

species (Larmola et al., 2014; Vile et al., 2014).  However, over 70 % of the ground cover in our rich fen site is

composed of brown mosses (Churchill, 2011), only some of which have been shown to fix N when exposed to

enough light (Basilier, 1979). Therefore, moss-based $N_2$ fixation may play a role in the N dynamics of the rich fen.

High N inputs could also result from inflows of N-rich surface or ground water.  Wetlands in the Tanana River

floodplain can be influenced by both surface runoff and river-based groundwater, as evidenced by $Ca^{++}$ values

(Racine and Walters, 1994). All ecosystems along the gradient, with the exception of the black spruce forest, have

been known to experience flooding during years of very high precipitation, with these flooding events dependent

on the behaviour of the Tanana River. During one of such events, Wyatt et al. (2011) found that dissolved inorganic

N (DIN) at our rich fen site peaked post-flood at ~0.50 mg $L^{-1}$. Dissolved organic N (DON) at this site has been

measured from ~ 0.86 – 1.42 mg $L^{-1}$ (Kane et al., 2010). While these DIN and DON concentrations are not

uncommon for a northern peatland (Limpens et al., 2006), the hydrologic connection between the fen and river is

undoubtedly important to the total N budget of the wetland. In addition, long-term influences such as disturbance

likely play an important role in N cycling (see below for more discussion regarding the influence of disturbance).

The higher long-term C accumulation rate for the rich fen compared to the other ecosystems suggests that

long-term C cycling is fundamentally different in the rich fen. The rich fen has significant deeper organic soil (91 cm

vs 30 cm or less for the other ecosystems). Mechanisms for C sequestration within this soil could be related to (1)

higher inputs into deep soil, from processes such as rooting, (2) less decomposable substrates, which in turn

reduces C losses, and/or (3) environmental conditions (i.e., soil temperature, oxygen availability) that reduce

decomposition losses. First, we examined rooting depth for each of the ecosystems. Descriptions of the rich fen

soil cores (Manies et al., 2016) show that live roots are found throughout the 90 cm organic soil profile, which is

significantly deeper than the other four ecosystems (Table 1). Therefore, input of C into the deep soil from roots is one possible mechanism for the larger amount of long-term C found at the rich fen. Next, we examined the C chemistry, or "quality", based on the organic soil C:N (Schädel et al., 2014). Lower C:N indicates substrate that has undergone more decomposition and, therefore, would likely be comprised of more recalcitrant material. A

comparison of surface C:N (< 20 cm) shows that the fen system has lower C: N than the black spruce or shrub ecosystems, but similar values to the grass and sedge ecosystems (Figure S1). This same pattern holds true for deeper soil layers (> 20 cm; Figure S1, note that the sedge site does not have organic soil deeper than 20 cm). More decomposable material could also be reflected in higher ER rates. However, McConnell et al. (2013) found that ER at the black spruce, shrub, and rich fen sites were statistically similar. Therefore, differences in

decomposable substrates likely do not play an important role in supporting deep soil C storage at the rich fen. Finally, we examined differences in environmental conditions, such as temperature and oxygen availability between the fen and other sites. Colder soil temperatures at depth at the rich fen could create slower rates of C cycling due to thermal protection. However, the rich fen site has warmer summer and annual soil temperatures at both 10 and 25 cm (Table 1), and the soil temperature is above freezing even at depth (there is no shallow

permafrost at this site). Therefore, preservation of C by thermal protection likely is not a contributor to the large amount of organic soil in this ecosystem. Another mechanism for reducing rates of C cycling is oxygen availability. McConnell et al. (2013) found lower $Q_{10}$ values at the rich fen, indicating less temperature sensitivity. Instead, with the shallowest water table (Table 1), it is thought that oxygen availability plays a dominant role in the protection of deep C at the rich fen (McConnell et al., 2013). Using average annual growth rates for the last 60 years, we found

that surface organics at rich fen become submerged in two decades, while it takes the surface material of the other ecosystems 40-90 years to reach the water table. Therefore, the rich fen organics are exposed to oxygen limiting conditions much more quickly than the other ecosystems.

Long-term C and N accumulation rates are also impacted by long-term factors, such as disturbance. The main disturbance in the boreal region is fire (Zoltai et al., 1998; Turetsky et al., 2011), which impacts the boreal C and N

cycles directly through emissions and indirectly via decreasing albedo (Ueyama et al., 2014), removing insulating organic soil layers (Pastick et al., 2014), and decreasing soil moisture (Carrasco et al., 2006), all of which impact decomposition rates. Because the rich fen has a shallower water table than the other ecosystems (Table 1), this

ecosystem is less likely to burn (Zoltai et al., 1998; Camill et al., 2009; Harden et al., 2000), even in dry years, and less severely if it does burn (Camill et al., 2009; Harden et al., 2000). Therefore, while the other ecosystems likely experienced many fires over the last several millennia, these fire events had a much smaller, if any, impact on C and N loss from the rich fen.

Decadal and long-term C accumulation rates can be used to constrain C accumulation rates as measured by eddy covariance flux towers. Euskirchen et al. (2014) examined annual C accumulation rates in 2012 and 2013 at this same rich fen location and found C accumulation rates of 36 and 127 gC m$^{-2}$ yr$^{-1}$, respectively. By comparison, our C accumulation rates ranged from 76 and 44 gC m$^{-2}$ yr$^{-1}$ (short- and long-term rates, respectively). The tower based net ecosystem exchange (NEE) in 2013 is 1.5 and 3 times higher than the decadal and long-term C accumulation rates found in this study, respectively, while the 2012 NEE rate is lower than both rates. As our decadal rates are averaged over the last six decades, this discrepancy suggests that the large C loss values Euskirchen et al. (2014) found in 2013 cannot be sustained over decades. Interannual variations in NEE for boreal systems are influenced by the length of the snow free season, soil temperature, light limitation (i.e., cloudiness), and changes in water table (Baldocchi, 2008). If tower measurements were continued over a longer time period we would expect high variability in annual NEE values and those values to be based on that year's weather conditions. Based on our decadal C accumulation rates years of high net C accumulation, like 2013, should be balanced out with years of net loss or low C accumulation to equal decadal rates from core profiles.

It is important to note that our sites are located close to the Tanana River and thus our findings are may be more indicative of locations where the groundwater can be influenced by river water. We also found a high level of within-ecosystem variability, with coefficients of variability of up to 60%. This variability is likely due microsite variability in surface vegetation, microtopography, and soil characteristics such as porosity, all which influence C and N cycling, and thus, accumulation rates. This variability limited our ability to make inferences about soil C and N accumulation rates between the four non-fen ecosystems. We also acknowledge that there is uncertainty associated with both dating techniques used in this study. Downward transport of [210]Pb could make the ages presented here appear younger than the actual age of the soil horizon. There are also potential uncertainties with [14]C ages due to the movement of younger atmospheric C into the soil through roots or fungi and the uptake of C from non-atmospheric sources (Bauer et al., 2009). To minimize these factors, future researchers could improve

upon our methods by increasing the number of soil cores, having higher resolution for soil horizons, and studying the possibility of $^{210}$Pb downwash using $^{7}$Be (Hansson et al., 2014). Regardless of the high within-ecosystem variability and potential accuracy of ages, we found significant differences in the long-term C and N accumulation rates of the rich fen in comparison to the other four ecosystems studied.

Future changes to Interior Alaska's climate are likely to affect C and N accumulation rates of the ecosystems

studied here differently. Increases in air temperatures (Hinzman et al., 2005) are likely to increase ER at the black spruce, shrub, grass and sedge ecosystems, based on findings by McConnell et al. (2013). This change will, in turn, reduce the decadal C accumulation rates of these ecosystems. However, climate induced shifts from vegetation from one ecosystem type to another among the four similar ecosystems should not impact either short- or long-term C accumulation rates, as we found similar rates among these four ecosystem types. Therefore, shifts between

these ecosystem types likely should not impact the regional C budget. This statement assumes, however, that any changes in climate influence the balance between C inputs and losses equally among ecosystems. Projected increases in fire severity and frequency (Turetsky et al., 2011) will also impact C accumulation rates, especially on the long-term. In contrast, rich fens are more likely to sustain their C and N accumulation rates as long as water tables are maintained as this high water table appears to diminish decomposition and reduce disturbance, thereby

helping the rich fen maintain its C and N stocks. However, the magnitude of the rate can be expected to be quite variable from year-to-year (Euskirchen et al., 2014; Baldocchi, 2008). The C and N balance of rich fens are like to be significantly impacted only if there are dramatic drops in water table (Waddington et al., 2014), which would require large changes to both the precipitation regime and subsurface hydrology (i.e., input sources of water), thereby increasing the susceptibility of the rich fen to wildfire and decreasing the zone of anoxic conditions, both

of which are important in maintain the large C and N stocks of this site.

**5 Conclusions**

This study provides C and N accumulation rates for a variety of northern ecosystems, many of which previously had little or no data available. Knowing rates of C and N accumulation in these five ecosystems will aid in the understanding of and ability to model their C & N cycles. For example, the overall C balance for four of the

five ecosystems were similar, even though inputs and losses are different, despite differences in dominant

vegetation, presence or absence of near-surface permafrost, and depth to water table. The significantly higher long-term C & N accumulation rates at the rich fen support the idea that long-term biogeochemical cycling in this ecosystem is different. We hypothesize that the black spruce, shrub, tussock grass, and sedge ecosystems experience more wildfires than the rich fen site, reducing their ability to preserve C and N over the long-term.

Additionally, C cycling in the rich fen ecosystem appears to be driven by different biogeochemical processes (such as lower oxygen availability) which results in the annual C balance of the rich fen more likely being a net C sink, thereby increasing long-term C accumulation rates. Climate change may increase rates of disturbance and soil temperatures for the non-rich fen ecosystems, impacting C and N accumulation rates. However, shifts from one ecosystem type to another among these four ecosystems would not impact regional C budgets. Our data also

suggest that climate change is less likely to significantly impact C budgets at the rich fen, as large changes in rich fen C accumulation rates would only occur if there is a dramatic drop in water table, which would require large changes to both the precipitation regime and subsurface hydrology.

**6 Acknowledgements**

We thank the Bonanza Creek Long-term Ecological Research program for granting us access to these research

sites. Their personnel, especially Jamie Hollingsworth, have been instrumental in providing support for this research. Thank you to Lee Pruett, Renata Mendieta, and Pedro Rodriguez for assisting with core collection, sample processing, or analyzing samples. We also thank Claire Treat, M. Braakhekke, and an anonymous reviewer for providing helpful comments on an earlier version of this manuscript. Funding for this work was provided by the U.S. Geological Survey Climate Research and Development program and the National Science Foundation (DEB-

0425328). The Bonanza Creek Long-term Ecological Research program is funded jointly by NSF (DEB-0620579) and the USDA Forest Service Pacific Northwest Research Program (PNW01-JV11261952-231).

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

**Table 1.** Site biological, physical, and chemical information. Depth of organic soil, based on three soil cores, are averages with standard deviations. July temperatures are averaged for 2005-2011. Water table depth from measurements after July 15 for the years 2005-2008.

| | Black spruce | Shrub | Tussock grass | Sedge | Rich fen |
|---|---|---|---|---|---|
| Dominant vegetation | *Ledum groendlandicum*, *Vaccinium caespitosum*, Feathermoss | *Salix* spp., *Betula* spp., *Chamaedaphne calyculata, Calamgrostis canadensis* | *Calamgrostis canadensis, Drepanocladus* spp. | *Carex atherodes* | *Drepanocladus* spp., *Sphagnum* spp., *Carex atherodes* |
| Depth of organic soil (cm) | 21 ± 2 | 30 ± 15 | 29 ± 22 | 16 ± 1 | 91 ± 12 |
| Shallow permafrost (<1 m) | yes | yes | no | no | no |
| Avg. July temperature at 10 cm (°C) [a] | 8.3 ± 2.3 | 5.7 ± 0.9 | 5.7 ± 2.3 | 9.1 ± 3.0 | 15.8 ± 5.2 |
| Avg. July temperature at 25 cm (°C) [a] | 2.0 ± 0.5 | 3.6 ± 0.7 | 5.1 ± 2.4 | 7.9 ± 3.1 | 11.2 ± 1.7 |
| Avg. annual temperature at 25 cm (°C) [b] | -0.03 ± 1.5 | -1.5 ± 4.0 | -1.3 ± 5.4 | -0.03 ± 5.3 | 2.1 ± 5.0 |
| Water table depth (cm) | 34 ± 6 | 12 ± 7 | 15 ± 13 | 11 ± 12 | 5 ± 11 |
| Soil moisture (% VMC at 5 cm) [b] | 15 ± 3 | 57 ± 8 | 66 ± 8 | 72 ± 7 | 84 ± 2 |

[a]2005-2011  [b]McConnell et al.

**Table 2.** Site C (g/m$^2$), N (g/m$^2$), and Unsupported $^{210}$Pb (dpm/cm$^2$) storage data. Unsupported $^{210}$Pb inventories represent the total atmospheric input of $^{210}$Pb to that site. Data are averages of three cores with standard deviations. Different letters after values indicate that the values among ecosystems are statistically different based on the Tukey Honest Significant Difference test.

| | Black spruce | Shrub | Tussock grass | Sedge | Rich fen |
|---|---|---|---|---|---|
| C storage in organic soil (g/m$^2$) | 6460$^a$ ± 940 | 14140$^a$ ± 4850 | 13950$^a$ ± 9130 | 7930$^a$ ± 1930 | 61500$^b$ ± 7290 |
| N storage in organic soil (g/m2) | 170$^a$ ± 20 | 700$^{ab}$ ± 150 | 940$^b$ ± 540 | 610$^a$ ± 120 | 3690$^c$ ± 190 |
| Unsupported $^{210}$Pb (dpm/cm$^2$) | 12.7$^a$ ± 5.6 | 10.7$^a$ ± 5.3 | 14.6$^a$ ± 2.7 | 10.2$^a$ ± 3.9 | 10.2$^a$ ± 1.3 |

**Table 3.** Decadal (< 60 yrs) and long-term (780-1400 yrs) C and N accumulation rates (g m$^{-2}$ yr$^{-1}$) with their standard deviations. Accumulation rates were determined by averaging values calculated for each individual soil profile by ecosystem type. Different letters indicate significant differences among ecosystems for that accumulation rate, based on Tukey Honest Significant Difference test.

| | Black spruce | Shrub | Tussock grass | Sedge | Rich fen |
|---|---|---|---|---|---|
| Decadal C accumulation rate (gC m$^{-2}$ yr$^{-1}$) | 59$^a$ ± 13 | 127$^a$ ± 73 | - | 73$^a$ ± 9 | 76$^a$ ± 9 |
| Long-term C accumulation rate (gC m$^{-2}$ yr$^{-1}$) | 8$^a$ ± 1 | 18$^a$ ± 6 | 18$^a$ ± 12 | 10$^a$ ± 2 | 44$^b$ ± 5 |
| Short:long C ratio | 7.1 | 7 | - | 7.2 | 1.7 |
| Decadal N accumulation rate (gN m$^{-2}$ yr$^{-1}$) | 1.4$^a$ ± 0.2 | 3.6$^{ab}$ ± 1.7 | - | 5.6$^b$ ± 0.6 | 4.6$^b$ ± 0.5 |
| Long-term N accumulation rate (gN m$^{-2}$ yr$^{-1}$) | 0.22$^a$ ± 0.03 | 0.90$^{ab}$ ± 0.19 | 1.20$^b$ ± 0.69 | 0.79$^a$ ± 0.16 | 2.66$^c$ ± 0.14 |

**Figure 1.** Comparison of $^{210}$Pb and $^{14}$C ages for depth increments where both analyses are available. The material dated for $^{14}$C ages was deciduous leaf fragments (shrub samples), seeds (sedge sample), or moss leaves and seeds (rich fen samples). The $^{210}$Pb values listed include estimated error.

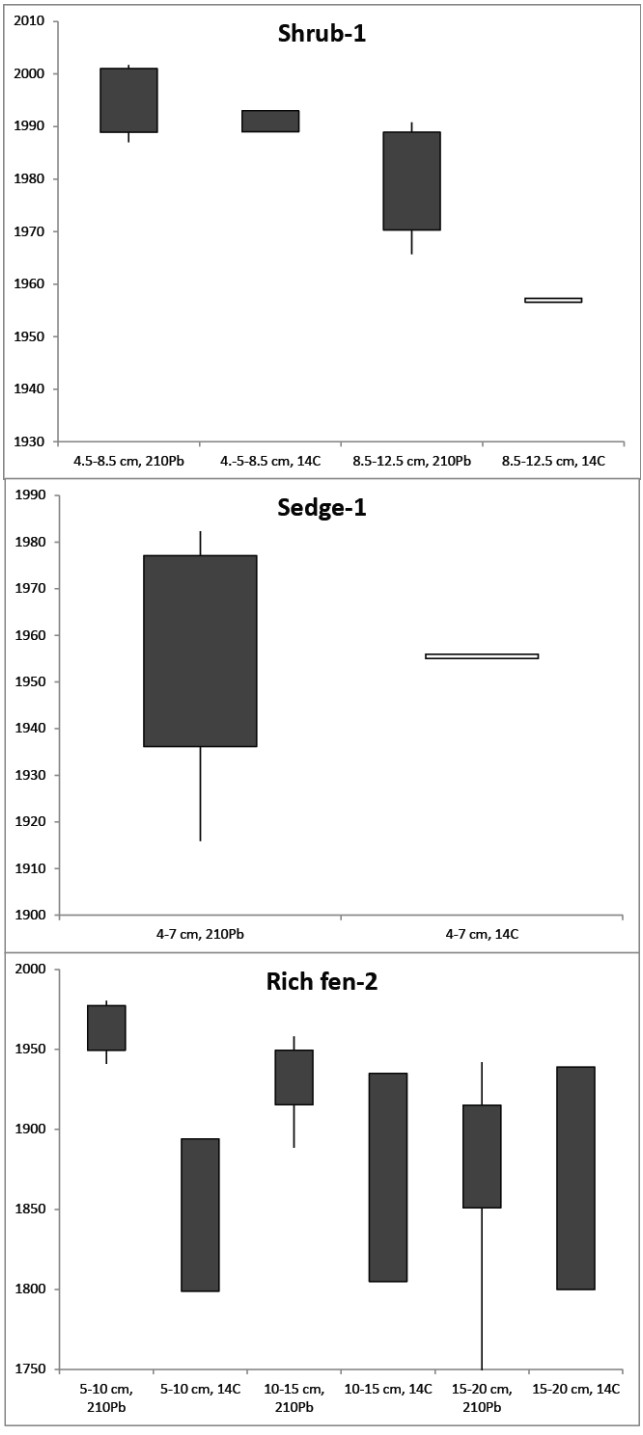

**Figure 2.** A comparison of organic soil C accumulation rates for fen and black spruce systems for this study (open symbols) and values found in the literature (solid symbols). Rates were calculated over many different time-spans (annual to millennial). Errors (where available) are standard deviations. Literature values from Aurela et al. (2004), Aurela et al. (2009), Bauer et al. (2009), Camill et al. (2009), Dunn et al. (2007), Euskirchen et al. (2014), Harden et al. (2012), Mathijssen et al. (2014), Oksanen et al. (2006), Turunen et al. (2002), Trumbore and Harden (1997), Yu et al. (2003) and Yu et al. (2013).

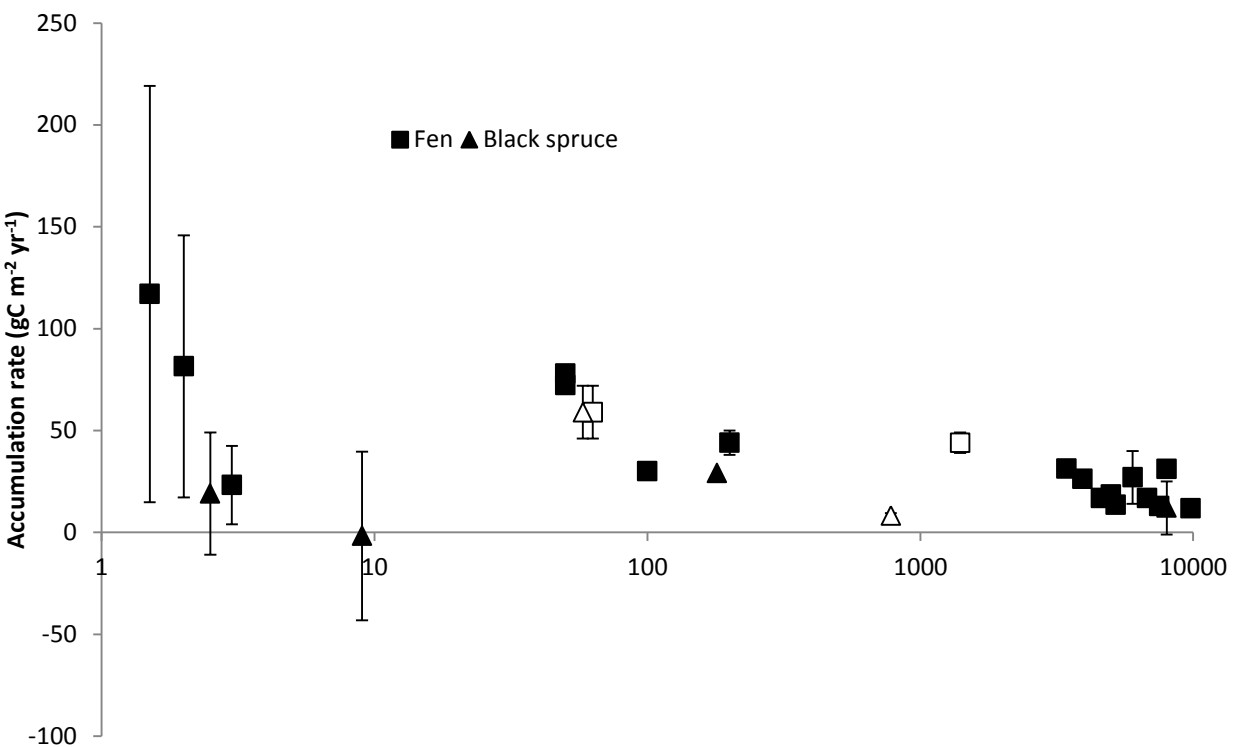

**Table S1.** [210]Pb ages for surface horizons for each profile, using both the CRS and CF:CS models. Errors are only shown for the CRS model as this is the model used in this study. Dates within square brackets have errors larger than the age difference between that horizon's age and the age of the intervals above, and therefore not considered valid. Depths with asterisks are mineral soil. BZBS = black spruce, BZWB = shrub, BZGR = tussock grass, BZEC = sedge, and BZRM = rich fen.

| Sample | Basal Depth (cm) | [210]Pb (dpm g$^{-1}$) | +/- | [226]Ra (dpm g$^{-1}$) | +/- | Dry mass (g cm$^{-2}$) | CRS basal date | +/- (yrs) | CF:CS basal date |
|---|---|---|---|---|---|---|---|---|---|
| BZBS 4 | 1 | 26.1 | 2.0 | 2.6 | 0.3 | 0.01 | 2005 | 0.1 | 2005 |
|  | 5 | 24.1 | 1.1 | 0.7 | 0.1 | 0.04 | 2002 | 0.3 | 2002 |
|  | 10 | 18.7 | 1.2 | 0.0 | 0.1 | 0.15 | 1990 | 1.4 | 1992 |
|  | 15 | 8.7 | 0.8 | 0.5 | 0.1 | 0.50 | 1956 | 7 | 1956 |
|  | 19 | 4.7 | 1.2 | 1.2 | 0.2 | 0.32 | 1928 | 20 | 1933 |
|  | 22 | 2.4 | 0.6 | 0.9 | 0.1 | 0.48 | [1863] | [144] | 1898 |
| BZBS 5 | 2 | 22.2 | 1.6 | 1.8 | 0.2 | 0.02 | 2005 | 0.2 | 2004 |
|  | 5 | 23.0 | 1.2 | 0.4 | 0.1 | 0.12 | 1992 | 1.2 | 1992 |
|  | 12.5 | 11.5 | 0.9 | 0.5 | 0.1 | 0.45 | 1945 | 10.2 | 1949 |
|  | 16.5 | 2.9 | 1.0 | 0.9 | 0.2 | 0.44 | 1881 | 112 | 1906 |
| BZBS 6 | 1 | 22.6 | 1.6 | 2.0 | 0.3 | 0.01 | 2006 | 0.0 | 2005 |
|  | 3 | 37.6 | 1.7 | 1.4 | 0.1 | 0.02 | 2004 | 0.1 | 2004 |
|  | 6 | 38.4 | 2.4 | 1.2 | 0.3 | 0.06 | 2000 | 0.4 | 1999 |
|  | 9 | 10.4 | 0.8 | 0.5 | 0.1 | 0.21 | 1995 | 0.6 | 1984 |
|  | 16 | 19.4 | 1.2 | 0.9 | 0.1 | 0.70 | 1936 | 9 | 1932 |
|  | 18 | 1.9 | 0.3 | 0.6 | 0.1 | 0.34 | [1871] | [299] | 1907 |
| BZWB 1 | 2 | 14.8 | 2.1 | 1.8 | 0.4 | 0.02 | 2005 | 0.2 | 2005 |
|  | 4.5 | 13.6 | 1.5 | 0.9 | 0.3 | 0.10 | 2001 | 0.7 | 2002 |
|  | 8.5 | 8.9 | 0.8 | 0.5 | 0.1 | 0.32 | 1988 | 1.9 | 1990 |
|  | 12.5 | 6.1 | 0.7 | 0.4 | 0.1 | 0.44 | 1970 | 4.6 | 1973 |
|  | 17.5 | 3.2 | 0.6 | 0.5 | 0.1 | 0.65 | 1946 | 12 | 1949 |
|  | 22.5 | 2.3 | 0.5 | 1.1 | 0.1 | 1.20 | [1881] | [91] | 1905 |
| BZWB 2 | 1 | 19.0 | 1.3 | 0.4 | 0.2 | 0.03 | 2003 | 0.4 | 2004 |
|  | 6 | 8.2 | 1.0 | 0.9 | 0.2 | 0.45 | 1975 | 6.0 | 1975 |
|  | 9 | 5.0 | 0.6 | 0.7 | 0.1 | 0.33 | 1941 | 17 | 1953 |
|  | 12 | 3.4 | 0.5 | 0.9 | 0.1 | 0.27 | [1877] | [129] | 1936 |

| | | | | | | | | | |
|---|---|---|---|---|---|---|---|---|---|
| BZWB 3 | 1 | 22.0 | 1.9 | 3.6 | 0.3 | 0.04 | 2005 | 0.2 | n.d. |
| | 6 | 6.8 | 0.8 | 0.8 | 0.1 | 0.55 | 1998 | 1.0 | n.d. |
| | 11 | 2.4 | 0.7 | 0.4 | 0.1 | 0.75 | 1994 | 1.7 | n.d. |
| | 16 | 3.8 | 0.8 | 0.8 | 0.2 | 0.40 | 1991 | 2.0 | n.d. |
| | 21 | 7.5 | 0.9 | 0.8 | 0.2 | 0.30 | 1985 | 2.2 | n.d. |
| | 26 | 6.3 | 0.9 | 0.4 | 0.2 | 0.40 | 1976 | 2.9 | n.d. |
| | 31 | 7.9 | 0.9 | 0.1 | 0.2 | 0.50 | 1951 | 7.0 | n.d. |
| | 36 | 8.1 | 0.9 | 0.3 | 0.2 | 0.40 | 1867 | 50 | n.d. |
| BZGR 1 | 3 | 12.9 | 0.8 | 0.6 | 0.1 | 0.21 | 1999 | 0.8 | 2001 |
| | 7 | 4.9 | 0.5 | 0.5 | 0.1 | 0.32 | 1994 | 1.1 | 1994 |
| | 11.5 | 5.0 | 0.6 | 0.8 | 0.1 | 1.22 | 1968 | 6.1 | 1966 |
| | 14 | 4.8 | 0.4 | 1.3 | 0.1 | 0.73 | 1935 | 16.9 | 1950 |
| | 18* | 2.2 | 0.3 | 1.4 | 0.1 | 1.60 | [1871] | [131.4] | 1914 |
| BZGR 2 | 5 | 8.0 | 0.8 | 0.9 | 0.1 | 0.40 | 2000 | 0.7 | 2001 |
| | 8 | 7.6 | 0.7 | 0.7 | 0.1 | 0.51 | 1992 | 1.3 | 1994 |
| | 10 | 5.3 | 0.6 | 0.8 | 0.1 | 0.36 | 1987 | 1.5 | 1989 |
| | 12 | 3.0 | 0.5 | 0.5 | 0.1 | 0.40 | 1983 | 1.6 | 1983 |
| | 14 | 5.2 | 0.6 | 0.7 | 0.1 | 0.44 | 1975 | 2.2 | 1977 |
| | 16 | 4.9 | 0.4 | 1.0 | 0.1 | 0.66 | 1960 | 3.5 | 1968 |
| | 18 | 3.0 | 0.4 | 1.6 | 0.1 | 0.96 | 1943 | 6.5 | 1955 |
| | 20* | 3.0 | 0.4 | 1.6 | 0.1 | 1.20 | 1909 | 23.4 | 1938 |
| | 22* | 2.4 | 0.4 | 1.7 | 0.1 | 1.12 | [1844] | [177] | 1923 |
| BZGR 3 | 5.5 | 6.9 | 0.6 | 0.3 | 0.1 | 0.39 | 1990 | 1.2 | 1989 |
| | 16 | 5.3 | 0.8 | 1.0 | 0.2 | 0.84 | 1986 | 3.7 | 1983. |
| | 30 | 4.5 | 0.7 | 0.2 | 0.1 | 0.84 | 1964 | 9.1 | 1967 |
| | 35 | 3.2 | 0.5 | 0.3 | 0.1 | 0.35 | 1952 | 12 | 1961 |
| | 41 | 1.3 | 0.5 | 0.4 | 0.1 | 0.78 | 1941 | 17 | 1946 |
| | 54* | 1.1 | 0.5 | 0.5 | 0.1 | 2.73 | [1877] | [156] | 1895 |
| BZEC 1 | 4 | 7.7 | 0.8 | 1.0 | 0.1 | 0.52 | 1978 | 5.0 | 1977 |
| | 7 | 3.8 | 0.5 | 0.7 | 0.1 | 0.57 | 1937 | 20 | 1946 |
| | 9 | 2.0 | 0.3 | 0.9 | 0.1 | 0.44 | [1872] | [166] | 1922 |
| BZEC 2 | 3 | 14.6 | 1.0 | 0.4 | 0.1 | 0.27 | 1995 | 1.1 | 1995 |
| | 7 | 11.1 | 0.9 | 1.1 | 0.1 | 0.56 | 1967 | 5.0 | 1972 |
| | 16 | 3.2 | 0.4 | 0.9 | 0.1 | 1.53 | 1902 | 38 | 1910 |

| | | | | | | | | | |
|---|---|---|---|---|---|---|---|---|---|
| BZEC 3 | 4 | 12.3 | 1.0 | 1.2 | 0.2 | 0.20 | 2001 | 0.6 | 2001 |
| | 7 | 12.5 | 0.9 | 1.0 | 0.1 | 0.48 | 1982 | 2.4 | 1988 |
| | 11 | 12.0 | 0.7 | 0.4 | 0.1 | 0.28 | 1960 | 4.7 | 1980 |
| | 15.5 | 4.7 | 0.7 | 1.0 | 0.1 | 0.90 | 1896 | 40 | 1956 |
| BZMR 1 | 2 | 5.6 | 0.4 | 0.1 | 0.1 | 0.12 | 2004 | 0.2 | 2001 |
| | 5 | 18.1 | 1.2 | 0.4 | 0.1 | 0.21 | 1990 | 1.3 | 1992 |
| | 10 | 9.3 | 0.7 | 0.4 | 0.1 | 0.35 | 1972 | 2.9 | 1978 |
| | 18 | 4.1 | 0.5 | 0.6 | 0.1 | 0.80 | 1933 | 13 | 1944 |
| | 32 | 1.0 | 0.6 | 0.4 | 0.1 | 1.54 | [1869] | [127] | 1880 |
| BZMR 2 | 2.5 | 16.3 | 0.8 | 0.9 | 0.1 | 0.35 | 1977 | 3 | 1982 |
| | 10 | 4.7 | 0.6 | 0.4 | 0.1 | 0.45 | 1950.1 | 8.4 | 1951 |
| | 15 | 3.0 | 0.4 | 1.1 | 0.1 | 0.50 | 1915.2 | 27 | 1917 |
| | 20 | 1.3 | 0.4 | 0.5 | 0.1 | 0.55 | [1851.0] | [208] | 1880 |
| BZMR 3 | 3 | n.d | - | n.d | - | n.d. | 2001.7 | 0.2 | 1989 |
| | 5 | 18.0 | 1.1 | 1.2 | 0.2 | 0.14 | 1991.6 | 1.0 | 1990 |
| | 7 | 15.0 | 1.9 | 1.3 | 0.3 | 0.16 | 1978.1 | 2.5 | 1980 |
| | 11 | 6.4 | 0.9 | 0.7 | 0.2 | 0.36 | 1956.9 | 5.8 | 1957 |
| | 13 | 5.9 | 1.0 | 1.3 | 0.2 | 0.20 | 1938.8 | 9.6 | 1945 |
| | 15 | 3.4 | 0.6 | 1.1 | 0.1 | 0.20 | 1922.7 | 15 | 1932 |
| | 17 | 2.3 | 0.5 | 1.1 | 0.1 | 0.22 | [1907.7] | [23] | 1918 |
| | 19 | 3.2 | 1.1 | 1.4 | 0.2 | 0.24 | [1843.5] | [193] | 1903 |

**Table S2.** [14]C raw and calibrated data. Negative values indicate ages younger than 1950 (present). These dates were calculated using CALIBomb with intercal13 and the NHZ1 bomb curve extension. All other dates were calibrated using CALIB 7.0, IntCal04 curve. Minimum and maximum ages are estimates using two sigma. Because ages are presented as cal BP, age at the time of sampling equals 2006-(1950-cal BP age).

| Profile | Depth (cm) | Lab Number | Description | Fraction Modern | FM Error | Age | Age error | Min age (cal BP) | Max age (cal BP) |
|---|---|---|---|---|---|---|---|---|---|
| Shrub-1 (BZWB 1) | 8.5 | WW9508 | Deciduous leaves of various sizes | 1.15 | 0.0041 | >Modern | - | -39.88 | -42.89 |
| | 12.5 | WW9509 | Deciduous leaves of various sizes | 1.05 | 0.0031 | >Modern | - | -6.58 | -7.30 |
| | 22.5 | WW9526 | Bulk soil with roots picked out | 0.97 | 0.0034 | 225 | 30 | 145 267 | 214 309 |
| | 27 | WW9527 | Bulk soil with roots picked out | 0.92 | 0.0027 | 695 | 25 | 646 | 683 |
| Sedge-1 (BZEC 1) | 7 | WW9522 | Seeds | 0.99 | 0.0052 | 55 | 45 | -5 | -5 |
| | 15 | WW9513 | Bulk soil with roots picked out | 0.91 | 0.0032 | 750 | 30 | 664 | 727 |
| | 17 | WW9514 | Bulk soil with roots picked out | 0.90 | 0.0026 | 865 | 45 | 725 | 800 |
| Rich Fen-2 (BZMR 2) | 10 | WW9523 | Moss leaves and seeds | 0.98 | 0.0042 | 130 | 35 | 56 172 | 151 278 |
| | 15 | WW9524 | Moss leaves and seeds | 0.99 | 0.0035 | 100 | 30 | 15 | 145 |
| | 20 | WW9525 | Moss leaves and seeds | 0.99 | 0.0043 | 115 | 35 | 11 | 150 |
| | 72 | WW9511 | Bulk soil with roots picked out | 0.86 | 0.0025 | 1225 | 25 | 1068 | 1187 |
| | 79 | WW9512 | Bulk soil with roots picked out | 0.84 | 0.0024 | 1435 | 25 | 1297 | 1371 |

**Table S3**. Results from the sensitivity analysis to determine the potential impact of different ages for the black spruce ecosystem. Long-term C accumulation rates using different ages for the black spruce ecosystem were compared to accumulation rates of the three other non-fen ecosystems (shrub, tussock grass, and sedge) and, separately, the rich fen ecosystem. Only when the black spruce ecosystem is 200 years old does its long-term C accumulation rate differ significantly from the non-fen ecosystems and become similar to the rich fen ecosystem. The long-term C accumulation rate for the black spruce ecosystem found in the manuscript is 8 +/- gC m$^{-2}$ yr$^{-1}$, based on an age of 780 yrs (see Section 3.2 for more information).

| Selected age of black spruce ecosystem | Long-term C accumulation rate using that age (gC m$^{-2}$ y$^{-1}$) | Are black spruce long-term C accumulation rates significantly different when compared to: | |
|---|---|---|---|
| | | Non-fen ecosystems (p-value)? | Rich-fen ecosystem (p-value)? |
| 200 yrs | 32 +/- 5 | Yes (0.031) | No (0.045) |
| 300 yrs | 22 +/- 3 | No (0.30) | Yes (0.003) |
| 500 yrs | 13 +/- 2 | No (0.45) | Yes (0.0006) |
| 1000 yrs | 6 +/- 1 | No (0.16) | Yes (0.0003) |
| 1400 yrs | 5 +/- 1 | No (0.11) | Yes (0.0002) |
| 1600 yrs | 4 +/- 1 | No (0.09) | Yes (0.0002) |

**Figure S1.** C:N relationships averaged by site for the surface organic soil (≤ 20 cm, panel 1) and deep organic soil (>20 cm, panel 2). Letters indicate significant differences (Tukey multiple comparison of means).

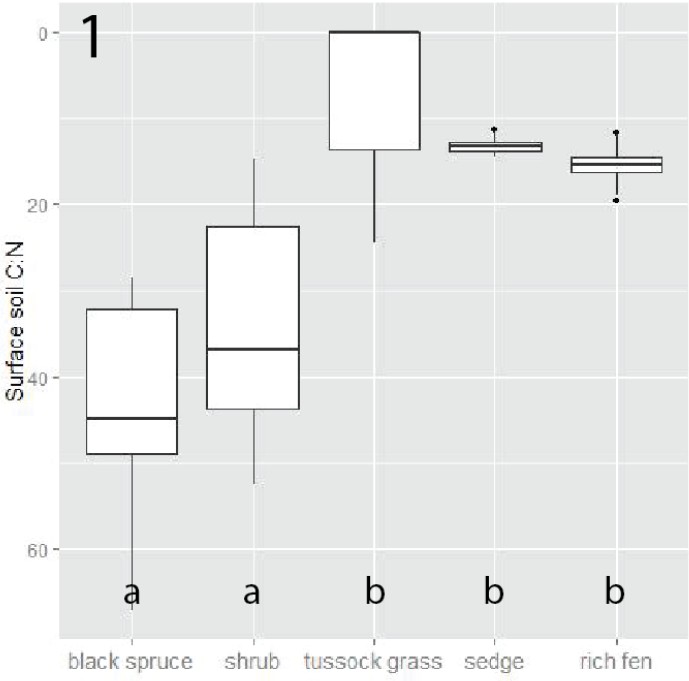

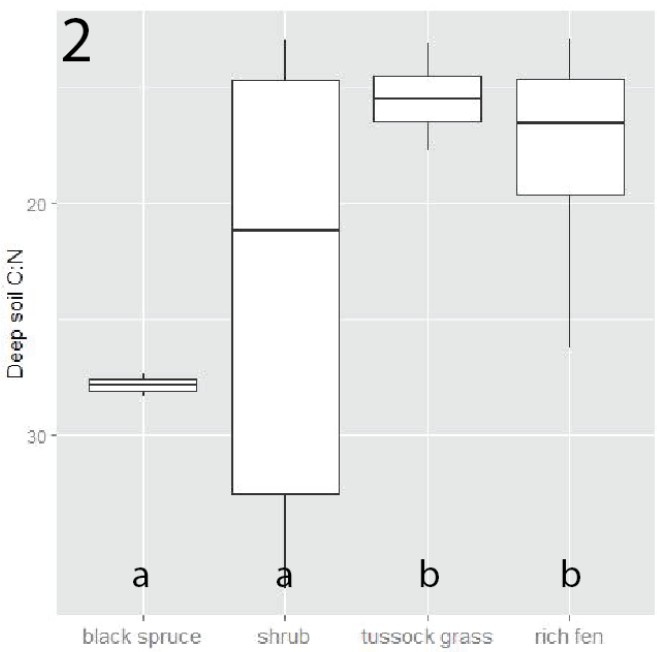