# Peer review of "Decadal and long-term boreal soil carbon and nitrogen sequestration rates across a variety of ecosystems"

_Biogeosciences, 2016_

## Referee Comment (RC1) · Anonymous Referee #1 · 22 Feb 2016

General comments

This study examined long-term and decadal carbon (C) and nitrogen (N) accumulation rates in five ecosystems, ranging from forest to grassland to fen, located along a hydrological gradient in an Alaskan floodplain. Such accumulation rate measurements are rare for some of the ecosystems and for N. The paper therefore presents novel data and addresses relevant scientific questions within the scope of Biogeosciences.

The organic soil profiles have been sampled at one site, so no real replicates, but the pro of this study is that the five ecosystems are close to each other, thus have developed under the same macro-environmental conditions (e.g. climate), allowing a comparison of the ecosystems. Unfortunately, for two ecosystems the age of the

organic soil profile could not be determined. For these two ecosystems the age was estimated assuming they have the same age as two other ecosystems, but the ground for this assumption is not clear. A description of the position of the five ecosystems in the landscape and their dynamics (vegetation succession, shifting mosaics?) would be helpful.

The main result was that the rich fen had higher long-term (700-1400 years) C and N accumulation rates than the other ecosystems, whereas the decadal (60 years) C accumulation rates were rather similar among the five ecosystems. Possible explanations for the high accumulation rates in rich fens are well discussed, but I would like to see more discussion on the N accumulation rates.

Overall, the paper presents interesting data, which deserve publication, but more attention should be given to the dynamics of these ecosystems in the floodplain landscape and to the N accumulation rates.

Specific comments

Title: I find temporal variation a bit misleading; it suggests that accumulation rates over multiple time periods have been compared, but the emphasis of the manuscript is on the (spatial) comparison of the five ecosystem types. Suggestion: Long-term and decadal carbon and nitrogen ...

L. 20-21: ... differences in the predominant mechanisms for nutrient cycling (for C) ... Please be more specific.

Introduction: very well written

L. 94-104: Please extend this description to include: are these ecosystems next to each other in the order given? What is their position in the floodplain landscape? What is the natural succession? How dynamic is the landscape?

L.102: Which sedge species is dominant in the "sedge" ecosystem?

L.110: Please give an indication of the thickness (... - ... cm) of the sampled soil horizons.

L.125: Please indicate how many soil horizon subsamples were measured per profile.

L.171-173: How likely is it that the black spruce and grass ecosystems have a similar age (= started developing at the same time) as the shrub and sedge ecosystems? For grass you may be safe, as it is in between the shrub and sedge ecosystem (assuming it is positioned in-between), but how about the black spruce forest? Without information on the development of these ecosystems I find this difficult to assess.

L.224-227: Phrase more carefully, there is uncertainty for two of these ecosystems (without independent age determination).

L.230-231: If soil temperature would be the driver for C cycling in these ecosystems I would expect the lowest (thus not the highest) C accumulation rate in the rich fen as it has (by far) the highest soil temperatures, promoting decomposition of the organic material.

L.241: Here I would like to see more discussion of the N accumulation rates. What is your explanation for the high N accumulation in the rich fen? The very high accumulation rate cannot originate from atmospheric N deposition alone; there must be other sources of nitrogen. Do the mosses in the rich fen have associations with N-fixing bacteria? Is there inflow of water relatively rich in nitrogen?

L.257-258: What about the Sphagnum mosses in the rich fen? Sphagnum mosses are known to be very recalcitrant to decomposition and could therefore contribute substantially to long-term C accumulation.

L.279: Can you support this discussion with observations of charcoal in the organic soil profiles?

Table 2: Could you add a line to the legend to explain what Unsupported 210Pb indicates/represents?

Table 2: A number is missing in the value for C storage in the rich fen

Table 3: Why not use the same layout as in Tables 1 and 2 with the ecosystems in columns.

Technical comments:

L.10: remove averaged

L.18-19: One decimal for the N accumulation rates is enough (the measurements were not that precise)

L.17: the highest instead of significantly higher

L.21: and instead of &

L.34: ... the net carbon balance of the boreal region?

L.114: remove a

L.191: had instead of has (results are written in past tense)

L.250: start a new paragraph with Second

L.286: C loss?

L.308: remove which appear

L.316: that that

---

## Referee Comment (RC2) · M. Braakhekke (Referee) · 22 Mar 2016

This paper presents a study on soil C and N accumulation rates on different time scales in boreal ecosystems. The authors used two techniques, 210Pb dating and 14C dating, to determine soil C & N accumulation rates on short and long time scale, respectively, for 5 ecosystems along a soil moisture gradient. Differences between the ecosystems are discussed in terms of factors that control C & N cycling. In general I find this an interesting and well written paper. Differences in soil C & N cycling in boreal ecosystems with different soil moisture regimes are highly relevant since climate change in high-latitude regions is likely to cause shifts in hydrological conditions, which will lead to vegetation change. It is also nice that the authors estimated accumulation rates on

two time scales.

General comments

I have some concerns about the use of the 210Pb dating method to determine short term accumulation rates. It seems to me that this approach hinges on the assumption that the effects of organic matter decomposition and vertical transport on the 210Pb profile are negligible. For both processes this may not be true. Decomposition is likely relevant, particularly for the dryer ecosystems. In fact, the authors acknowledge in the discussion that the accumulation rates is the result of the balance between input and decomposition. Significant loss of organic matter by decomposition would cause 210Pb to become more concentrated, resulting in underestimation of the age. In two cases the 210Pb age is significantly lower than the 14C age (Figure 1)–it seems to me that this could be explained by the effects of decomposition. With regard to vertical transport, the authors indicate that this may be relevant for the Tussock grass site (section 3.1) because of the occurrence of 210Pb in the mineral soil. However, the fact that 210Pb is not found in the mineral soil for the other sites is no assurance that vertical transport is not relevant there. It is good that the authors include 14C measurements for validation of the 210Pb ages. However, I think some more justification of the approach is appropriate. For example, based on previously published decomposition rates for similar soils the authors could estimate the effects of decomposition on the 210Pb concentration. Also, I think a honest discussion of the limitations and uncertainties in section 4 should be added.

Specific comments

- p 3/l 55-56: This sentence is not clear to me. What does "these ecosystems" refer to?

- p 5/l 116: please indicate the units of the mesh

- p 6/l 148: "Bulk peat samples" suggests that these measurements were only performed for the fen/bog soils but later text suggest that these measurements were done for all sites. Please clarify.

- p 6/l 154: "age of that profile": is that the age inferred from the 14C measurement of the "basal soil organic horizon"? p 7/l 174: I found this sentence somewhat confusing. It seems that the sample, including macrofossils, is homogenized, which is not the case, I assume.

- p 10/ l 235: please remove the closing parenthesis ")" at the end of the sentence of insert an opening parenthesis somewhere appropriate

- p 11/l 266: It is not clear to me how decreasing q10 values suggest that oxygen availability is a dominant factor for C preservation

- p 12/l 311: please add "of" after "many"

- Table 3: The short term accumulation rates were determined horizon-wise in the table single numbers are given. Are these averages over all horizons?

- Table 3: Please indicate what the superscript letters a,b,c mean. In the text (section 3.3) it is written that the decadal C accumulation rates are not significantly different between the sites, but this is not clear from the letter "a" in the table.

- Table S2 (supplement), caption: I assume you mean "younger" than 1950, not "older"

---

## Author Comment (AC1)

**Responses to reviewer #1**

*Suggestion to change the title*. The title has been adjusted to reflect your suggestions and now is "Decadal and long-term boreal soil in carbon and nitrogen sequestration rates across a variety of ecosystems".

*Please be more specific regarding the mechanisms in lines 20-21*. This information has been added: "Our results suggest that the controls on long-term C and N cycling at the rich fen is fundamentally different from the other ecosystems, likely due to differences in the predominant drivers of nutrient cycling (oxygen availability, for C) and reduced amounts of disturbance by fire (for C and N)"

**Introduction is well written. Thank you.**

*Please extend the descriptions of the ecosystems*. We have addressed this comment by adding the following text after these ecosystems are introduced (line 106): "This transect extends from the toe slope of an adjacent upland forest into a ~1.8 km2 fen complex. Although in the Tanana floodplain, the sites are ~1.5 km from the current location of the river and appear to be relatively stable since site initiation in 2005. These sites have also been a part of other studies, including examining controls on ecosystem respiration (McConnell et al., 2013), examining differences in the soil biotic community and their impact on soil C turnover (Waldrop et al., 2012), understanding how changing water table level impacts C cycling within the fen (Kane et al., 2013; Chivers et al., 2009), and using eddy covariance methods to calculate net ecosystem productivity (Euskirchen et al., 2014)."

Which sedge species is dominant in the sedge ecosystem? The text (line 104) now reads "4) a peatland dominated by emergent vegetation such as *Equisetum fluviatile* ("sedge"), ...".

Please give an indication of the thickness of the sample soil horizons. We have added a sentence to reflect this information (line 118): "Soil horizon thicknesses ranged between 2-14 cm, with 85% of samples having a thickness ≤5 cm."

How likely is it that the black spruce and grass ecosystems have a similar age as the shrub and sedge ecosystem? ... Without information on the development of these ecosystems I find this difficult to assess. We have added some information to help the reader understand why we feel that this assumption is justified (line 186): "We feel that this assumption is justified because a) the grass ecosystem lies between the shrub and sedge ecosystems, and b) the black spruce would need a dramatically different initiation age (<250 yrs) for the long-term C storage at this ecosystem to be significantly different than the other non-fen ecosystems, indicating that even if this age is not accurate for the black spruce ecosystem changes to this date would not change our results."

*Lines 224-227: Phrase more carefully, there is uncertainty for two of these ecosystems*. While there is uncertainty in the two undated ecosystems, we feel that due to the justification, as written above, this statement holds true.

If soil temperature would be a driver for C cycling in these ecosystems I would expect the lowest (thus not the highest) C accumulation rate in the rich fen as it has (by far) the highest soil temperatures, promoting

*decomposition of the organic material.* This statement would be true if temperature was the regulating factor for C at the rich fen. However, McConnell et al. found that temperature was not the main driver for ecosystem respiration (ER) at the rich fen site. Instead, it appears that the availability of oxygen is the primary factor in determining C cycling. The text that explains this can be found on line 300: "Another mechanism for reducing rates of C cycling is oxygen availability. McConnell et al. (2013) found lower Q10 values at the rich fen, indicating less temperature sensitivity. Instead, with the shallowest water table (Table 1), it is thought that oxygen availability plays a dominant role in the protection of deep C at the rich fen (McConnell et al., 2013)."

Here I would like to see more discussion of the N accumulation rates. What is your explanation for the high N accumulation in the rich fen? The very high accumulation rate cannot originate from atmospheric N deposition alone; there must be other sources of nitrogen. Do the mosses in the rich fen have associations with N-fixing bacteria? Is there inflow of water relatively rich in nitrogen? We had expanded the N paragraph greatly, examining several theories as to why the rich fen might have high N. In addition, the manuscript discusses the fact that reduced disturbance due to fire (line 307) would impact both C and N storage. The expanded paragraph (line 254) is as follows: "Nitrogen accumulation rates have been studied much less frequently than rates of C accumulation. The long-term N accumulation rate for the rich fen in this study (2.66 g N m-2 yr-1) is five times higher than the 0.5 g N m-2 yr- estimated by Loisel et al. (2014). There are several potential reasons for this discrepancy. First, Loisel et al. (2014) synthesized data from a wide range of peatland sites, including bogs, fens, and permafrost peatlands and thus included ecosystems with a broad spectrum of peat properties. In addition, Loisel et al. (2014) used time-dependent C:N ratios of 65 and 40 to assign % N values for their soil horizons, resulting in average % N values that never exceed 1.7 %. In contrast, the average % N value for our rich fen organic soil horizons was 2.4 %, resulting in an average C:N ratio of 17 (Fig S1). In general, our results support Treat et al. (2015), who showed that fen C:N ratios can be much lower than estimates used by Loisel et al. (2014), despite high variability (fen C:N averaging 29 +/- 15). Regardless, the amount of N within the rich fen ecosystem is relatively high. Reasons for this high N storage could include high rates of N inputs, either through high rates of biological N2 fixation or through high N concentrations in source water. The majority of studies on N fixation in peatlands have focused on Sphagnum species (Larmola et al., 2014; Vile et al., 2014). However, over 70 % of the ground cover in our rich fen site is composed of brown mosses (Churchill, 2011), some of which have been shown to fix N when exposed to enough light (Basilier, 1979). Therefore, moss-based N2 fixation may play a role in the N dynamics of the rich fen. High N inputs could also result from inflows of N-rich surface or ground water. Wetlands in the Tanana River floodplain are influenced by both surface runoff and river-based groundwater, as evidenced by Ca++ values (Racine and Walters, 1994). All ecosystems along the gradient, with the exception of the black spruce forest, have been known to experience flooding during years of very high precipitation, with these flooding events dependent on the behaviour of the Tanana River. During one of such events, Wyatt et al. (2011) found that dissolved inorganic N (DIN) at our rich fen site peaked post-flood at ~0.50 mg L-1. Dissolved organic N (DON) at this site has been measured from ~ 0.86 – 1.42 mg L-1 (Kane et al., 2010). While these DIN and DON concentrations are not uncommon for a northern peatland (Limpens et al., 2006), the hydrologic connection between the fen and river is undoubtedly important to the total N budget of the wetland."

What about the Sphagnum mosses in the rich fen? Sphagnum mosses are known to be very recalcitrant to decomposition and could therefore contribute substantially to long-term C accumulation. The fact that Sphagnum mosses do not comprise the majority of groundcover. This fact and the possible influence of mosses on our N results is now discussed in the paragraph above. The text regarding mosses can be found on line 265: "The majority of studies on N fixation in peatlands have focused on Sphagnum species (Larmola et al., 2014; Vile et al., 2014). However, over 70 % of the ground cover in our rich fen site is composed of brown mosses (Churchill, 2011), some of which have been shown to fix N when exposed to enough light (Basilier, 1979). Therefore, moss-based N2 fixation may play a role in the N dynamics of the rich fen."

*Can you support this discussion with observations of charcoal in the organic soil profiles*? Unfortunately, we only found one soil horizon with charcoal in it during our visual inspections. While there is likely much macroscopic charcoal within these samples, that examination is beyond the scope of this study.

*Table 2: Could you add a line to the legend to explain what Unsupported*210*Pb indicates/represents*? The following text has been added: "Unsupported 210Pb stocks represent the total atmospheric input of 210Pb to that site."

*Table 2: A number is missing in the value for C storage in the rich fen.* Thanks for catching this error. The number has now been corrected to  $61500 \text{ gC} / \text{m}^2$ .

*Table 3: Why not use the same layout as in Tables 1 and 2 with the ecosystems in columns.* Good suggestion. This change has been made.

---

## Author Comment (AC2)

**Responses to reviewer #2**

I have some concerns about the use of the 210Pb dating method to determine short term accumulation rates. It seems to me that this approach hinges on the assumption that the effects of organic matter decomposition and vertical transport on the 210Pb profile are negligible. For both processes this may not be true. Decomposition is likely relevant, particularly for the dryer ecosystems. In fact, the authors acknowledge in the discussion that the accumulation rates are the result of the balance between input and decomposition. Significant loss of organic matter by decomposition would cause 210Pb to become more concentrated, resulting in underestimation of the age. In two cases the 210Pb age is significantly lower than the 14C age (Figure 1)–it seems to me that this could be explained by the effects of decomposition. With regard to vertical transport, the authors indicate that this may be relevant for the Tussock grass site (section 3.1) because of the occurrence of 210Pb in the mineral soil. However, the fact that 210Pb is not found in the mineral soil for the other sites is no assurance that vertical transport is not relevant there. It is good that the authors include 14C measurements for validation of the 210Pb ages. However, I think some more justification of the approach is appropriate. For example, based on previously published decomposition rates for similar soils the authors could estimate the effects of decomposition on the 210Pb concentration.

The reviewer is correct that decomposition can influence 210Pb ages if you are modeling them on the basis of 210Pb activity per gram (dpm/g) vs depth (cm) since loss of mass by degradation will increase the 210Pb activity per gram. However, we have accounted for the influence of decomposition as well as compaction over time by modeling our 210Pb profiles on a drymass basis (e.g. vs cumulative dry mass, g/cm2) instead of depth. This approach derives accumulation rates from the activity of unsupported 210Pb within the entire volume of interval based on the bulk density, which also increases in response to compaction and organic matter degradation, thus accounting for both effects. We have made this point clear in the text (line 138) by adding: "To account for compaction and loss of mass due to organic matter decomposition, both methods modelled unsupported 210Pb as a function of cumulative dry mass (g/cm2), not depth (Appleby and Oldfield, 1992). Cumulative dry mass is the product of bulk density of the horizon (g/cm3) and the horizon thickness (cm)" Although the effects of decomposition and/or compaction are addressed with our methodology, we still need to considerw the effect of movement of 210Pb down the soil profile on age estimates. For this reason we submitted surface samples for 14C data, hoping to corroborate all 210Pb dates with 14C dates. This possibility is now explicitly addressed within the results section. As previously mentioned in the manuscript, adjusting the dates of the shrub ecosystem to the 14C ages does not impact our results. Therefore, we feel comfortable moving forward using the 210Pb dates. The new text (line 198) is as follows: "The younger 210Pb date for the 8.5 – 12.5 cm Shrub-1 horizon could indicate that there has been some movement of 210Pb within the soil profile, which has been known to occur with this dating technique (Turetsky et al., 2004). However, the 14C and 210Pb ages for the 4.5 – 8.5 cm horizon match well, which we would not expect if downward transport was a significant issue. In addition, adjusting our analyses to the 14C dates does not change our results. Therefore, we feel comfortable moving forward using the 210Pb age values."

*I think a honest discussion of the limitations and uncertainties in Section 4 should be added*. We have added a paragraph (line 329) discussing both the limitation and uncertainties of our data. It reads "It is

important to note that our sites are located close to the Tanana River and thus our findings are may be more indicative of locations where the groundwater can be influenced by river water. We also found a high level of within-ecosystem variability, with coefficients of variability of up to 60%. This variability is likely due microsite variability in surface vegetation, microtopography, and soil characteristics such as porosity, all influence which C and N cycling, and thus, accumulation rates. This variability limited our ability to make inferences about soil C and N accumulation rates between the four non-fen ecosystems. We also acknowledge that there is uncertainty associated with both dating techniques used in this study. Downward transport of 210Pb could make the ages presented here appear younger than the actual age of the soil horizon. There are also potential uncertainties with 14C ages due to the movement of younger atmospheric C into the soil through roots or fungi and the uptake of C from non-atmospheric sources (Bauer et al., 2009). To minimize these factors, future researchers could improve upon our methods by increasing the number of soil cores, having higher resolution for soil horizons, and studying the possibility of 210Pb downwash using 7Be (Hansson et al., 2014). Regardless of the high withinecosystem variability and potential accuracy of ages, we found significant differences in the long-term C and N accumulation rates of the rich fen in comparison to the other four ecosystems studied.

*p 3/l 55-56: This sentence is not clear to me. What does "these ecosystems" refer to?* Clarified to read "fen ecosystems" (line 57).

p 5/l 116: please indicate the units of the mesh. Added information that 60 mesh is 0.25 mm (line 126).

*p* 6/l 148: "Bulk peat samples" suggests that these measurements were only performed for the fen/bog soils but later text suggest that these measurements were done for all sites. Please clarify. Sentence rewritten to make it clear that it was soil, not peat, so all sites were included: "Additionally, bulk soil samples, with roots removed, were submitted..." (line 60).

*p* 6/l 154: "age of that profile": is that the age inferred from the 14C measurement of the "basal soil organic horizon"? The sentences (line 66) have been rewritten to clarify. " Long-term C accumulation rates were calculated as the amount of C within the organic soil profile divided by the 14C age of that ecosystem. Ecosystem age was calculated as the average of the minimum and maximum 14C calibrated ages (Suppl. Table S2)."

*p* 7/l 174: I found this sentence somewhat confusing. It seems that the sample, including macrofossils, is homogenized, which is not the case, I assume. Unfortunately, when we divided the soil cores into horizons we were not planning on sampling macrofossils. Therefore, the horizons are wider than usual and were homogenized when splitting them for analytical sampling and creating an archive. This means that macrofossils could have come from anywhere within the horizon (surface, middle, base). This fact is now explicitly stated in the methods section in two places: line 122) "Soils horizon samples were processed in several steps: first they were air dried (20-25 °C) and then homogenized. The samples were then split into two parts: an archive split and an analytical split. The analytical split was oven dried and then ground." and line 159) "We also dated macrofossils, obtained from several, homogenized soil horizons, using AMS radiocarbon measurements for comparison to 210Pb ages. (Suppl. Material S2)."

*p* 10/ l 235: please remove the closing parenthesis ")" at the end of the sentence or insert an opening parenthesis somewhere appropriate. Done.

*p* 11/l 266: It is not clear to me how decreasing q10 values suggest that oxygen availability is a dominant factor for C preservation. We have rewritten these sentences (line 300) to make them clearer. "Another mechanism for reducing rates of C cycling is oxygen availability. McConnell et al. (2013) found lower Q10 values at the rich fen, indicating less temperature sensitivity. Instead, with the shallowest water table (Table 1), it is thought that oxygen availability plays a dominant role in the protection of deep C at the rich fen (McConnell et al., 2013)."

**p 12/l 311: please add "of" after "many". Done**

Table 3: The short term accumulation rates were determined horizon-wise in the table single numbers are given. Are these averages over all horizons? We have added the following text to the table caption to clarify how these were calculated. "Accumulation rates were determined by averaging values calculated for each individual soil profile by ecosystem type."

Table 3: Please indicate what the superscript letters a,b,c mean. In the text (section 3.3) it is written that the decadal C accumulation rates are not significantly different between the sites, but this is not clear from the letter "a" in the table. We have clarified what the superscripts mean with the following text: "Different letters indicate significant differences among ecosystems for that accumulation rate, based on Tukey Honest Significant Difference test."

Table S2 (supplement), caption: I assume you mean "younger" than 1950, not "older". Changed.

---

## Author Comment (AC3)

[revised manuscript text omitted]

Age estimates from the base of the organic soil were similar for the shrub and sedge ecosystems (720 and 840 yrs cal BP). Because we did not have ages for the black spruce or grass ecosystem (due to sample size limitations)

185     we averaged the ages measured for the shrub and sedge systems and used this value as the age of formation for

all ecosystems with the exception of the rich fen. We feel that this assumption is justified because a) the grass ecosystem lies between the shrub and sedge ecosystems, and b) the black spruce would need a dramatically different initiation age (<250 yrs) for the long-term C storage at this ecosystem to be significantly different than the other non-fen ecosystems, indicating that even if this age is not accurate for the black spruce ecosystem changes to this date would not change our results. [14]C dating of basal organics shows the rich fen is older, approximately 1400 years old.

[revised manuscript text omitted]

---

## Author Comment (AC4)

[revised manuscript text omitted]

---

## Author Response (AR1)

Dear Dr. Zaehle,

Thank you very much for you suggestions on how to further improve our manuscript. We have uploaded a new version with changes made based on your suggestions. After a careful review we made sure that our manuscript either restates pertinent information, or refers the reader to the location of that information within the text (i.e., line 282). In addition, we have clarified our justification in assigning the ages used to calculate long-term C accumulation rates. This includes showing the results of our sensitivity analysis (new Table S3), which demonstrates that any uncertainty related to the age of the black spruce profile would not affect our results.

[Figure]

The updated text (line 183) and Table S3 are below. Please let me know if you have any additional questions.

Thank you,

Kristen Manies

Updated text: "14C dating of the basal organic soil layers provided information regarding the initiation of soil development. This approach shows that the rich fen is the oldest ecosystem, at approximately 1390 years old (Table S2). Age estimates for the shrub and sedge ecosystems ranged between 700 and 856 yrs cal BP. Unfortunately, we did not get ages for the black spruce or tussock grass ecosystem (due to sample size limitations). Therefore, for all ecosystems except the rich fen we used an initiation age of 780 yrs (the median of the two ages listed above). We justify this approach using the following logic. First, all of the ecosystems appear to be relatively stable and lay within ~300 m of each other, along an emergent landform that grades from the rich fen up to the black spruce forest. Therefore, all ecosystems along this gradient likely formed within several hundred years of each other. This assumption is supported by the fact that the sedge ecosystem is only ~100 years older than the shrub ecosystem. The grass ecosystem also lies between the shrub and sedge ecosystems along the gradient; therefore, its age of formation is likely similar to the values measured for these two ecosystems. Although the black spruce ecosystem lies at the end of the gradient, a sensitivity analysis demonstrates that a dramatically different initiation age would be needed to impact our results (Table S3). Therefore, even if 780 yrs is not accurate for the black spruce ecosystem, realistic variations in this value (+/- 400 years) would not change the outcome of our analyses."

Please also note the supplement to this comment:
http://www.biogeosciences-discuss.net/bg-2016-24/bg-2016-24-AC4-supplement.pdf

[revised manuscript text omitted]

---

## Author Response (AR2)

**Response to 2nd review**

**Report #1**

No comments

**Report #2**

Line 65: The authors mean that higher lignin:N ration has lower decomposition, right?

This wording has been changed: "Litter composed of more complex C compounds and/or higher lignin:N ratios can have lower decomposition rates and, therefore, lower rates of C loss and relative N retention."

Line 195: It seems that tit would be appropriate to mention the stand age of the black spruce, which would also indicate the time since stand replacing fire.

The fire return interval of our black spruce forest, based on literature values, is likely between 25 - 150 years\*. However, stand age has no influence on and is much younger than the age of ecosystem initiation, which is what is needed to determine long-term accumulation rates. Therefore, we did not measure this value.

\*Fryer, Janet L. 2014. Fire regimes of Alaskan black spruce communities. In: Fire Effects Information System, [Online]. U.S. Department of Agriculture, Forest Service, Rocky Mountain Research Station, Fire Sciences Laboratory (Producer). Available: http://www.fs.fed.us/database/feis/fire\_regimes/AK\_black\_spruce/all.html [2016, July 13].

Line 320: Was there any charcoal evidence indicating fires that could help corroborate this? A lack of any charcoal in the core descriptions from the rich fen would show this clearly.

We addressed the issue of charcoal evidence by adding the following text: "An examination of the soil horizon descriptions for these sites only found one incidence of charcoal being observed (Manies et al., 2016; shrub ecosystem, 12-17 cm). However, a macrofossil analysis of these or future cores could further support this hypothesis."

Line 335: remove "are"

Removed extra word.

Line 337: "...This variability is likely due [to] microsite variability"

Add the word 'to'.

Figure 1: Perhaps this would depict the comparison better as an xy plot (Pb vs. C) with correlation coefficients.

We adjusted this figure to be a xy plot and found it to be confusing, especially because each date has an upper and a lower range, which is difficult to interpret in the xy format. We would like to keep this figure as it is. To make the figure clearer we added some vertical lines to make it easier to delineate which samples should be compared.